# Chemically Modified Starches as Food Additives

**DOI:** 10.3390/molecules28227543

**Published:** 2023-11-11

**Authors:** Dorota Gałkowska, Kamila Kapuśniak, Lesław Juszczak

**Affiliations:** 1Department of Food Analysis and Evaluation of Food Quality, University of Agriculture in Krakow, Balicka 122, 30-149 Krakow, Poland; dorota.galkowska@urk.edu.pl; 2Department of Dietetics and Food Studies, Jan Dlugosz University in Czestochowa, Armii Krajowej 13/15, 42-200 Częstochowa, Poland; k.kapusniak@ujd.edu.pl

**Keywords:** starch, chemical modification, food additives, industrial use

## Abstract

Starch is a renewable and multifunctional polysaccharide biopolymer that is widely used both in the food industry and other areas of the economy. However, due to a number of undesirable properties in technological processes, it is subjected to various modifications. They improve its functional properties and enable the starch to be widely used in various industries. A modified starch is a natural starch that has been treated in a way that changes one or more of its initial physical and/or chemical properties. Chemical modification consists of the introduction of functional groups into starch molecules, which result in specific changes in the physicochemical and functional properties of starch preparations. The bases of chemical modifications of starch are oxidation, esterification or etherification reactions. In terms of functionality, modified preparations include cross-linked and stabilized starches. These starches have the status of allowed food additives, and their use is strictly regulated by relevant laws. Large-scale scientific research is aimed at developing new methods of starch modification, and the use of innovative technological solutions allows for an increasingly wider use of such preparations. This paper characterizes chemically modified starches used as food additives, including the requirements for such preparations and the directions of their practical application. Health-promoting aspects of the use of chemically modified starches concerning resistant starch type RS4, encapsulation of bioactive ingredients, starch fat substitutes, and carriers of microelements are also described. The topic of new trends in the use of chemically modified starches, including the production of biodegradable films, edible coatings, and nanomaterials, is also addressed.

## 1. Introduction

Progress in food processing and the continuing high demand for convenient and functional foods have resulted in increased use of allowed food additives in food processing. The use of these substances in accordance with current health and legal recommendations enables food producers to achieve the intended technological, quality, economic, and health effects. Due to their useful and functional values, food additives are essential components of many food products. Although they constantly raise health concerns among some consumers due to the increase in public awareness of the role and the legally regulated conditions of use of food additives, they are becoming more and more widely accepted [1].

Allowed additives are substances not consumed separately as food and are not typical food ingredients; allowed additives are intentionally used in the process of production, processing, preparation, packaging, transport, and storage to provide the intended or expected results in a food or in semi-finished products that are components of the food. Allowed additives may be used only when their use is compliant with current legal regulations and technologically justified and does not pose a threat to the health or life of the consumer. The basic legal acts in the European area regulating the use of food additives are Regulation (EC) No. 1333/2008 of the European Parliament and of the Council of 16 December 2008 on food additives [2] and Commission Regulation (EU) No. 1129/2011 of 11 November 2011 amending Annex II to Regulation (EC) No 1333/2008 of the European Parliament and of the Council by establishing a Union list of food additives [3].

As a fully renewable and multifunctional polysaccharide, starch is widely used in the food industry and other areas of the economy. It is obtained from various natural sources, including grains (cereals and pseudo-cereals), seeds (e.g., pulses), tubers, roots, stems, and leaves; starch from corn grains account for 75% of the world market production followed by starch from cassava (14%), wheat (7%), and potatoes (4%) [4,5,6]. Native starches, however, show some shortcomings limiting their wide use in complex food systems. These include low swelling capacity and poor solubility; limited thermal and mechanical stability and low general stability in freeze-thaw cycles; low clarity of gels and the tendency of starch in gels to undergo retrogradation, resulting in syneresis of gels [7,8,9,10]. The use of a specific type of modification eliminates these unfavourable properties and gives the starch new desirable functional features [11]. Modifications based on physical, chemical, enzymatic, or combined methods are used [12]. Modified starch is natural starch treated in a way that changes one or more of its initial physical or chemical properties [13]. The chemically modified starches are considered as food additives with the label of “modified starch” and an E-number coded according to the International Numbering System [14]. In contrast, starches modified by physical or enzymatic methods do not have the status of food additives, they are only food ingredients, unless the physical or enzymatic modification applied concerns already chemically modified starch, e.g., pregelatinized sodium octenyl succinate starch.

The advantages of chemically modified starches include the nature of the substrate for their production, i.e., the naturalness of the starch, its easy availability, renewability, biodegradability, and functionality. Moreover, the costs of producing chemically modified starches are acceptable. Chemical modification of starch allows obtaining derivatives with desired functional properties, intended for various branches of the food industry. Moreover, chemically modified starches can support the production of functional food, e.g., by introducing resistant starch, stabilizers, and starch-encapsulated bioactive compounds into recipes. Possible concerns about the health aspects of consuming food products containing chemically modified starches are not justified. The chemical reactions used to modify starch (oxidation or esterification) also occur in the human body during carbohydrate metabolism. Moreover, legal regulations regarding the use of specific modifying reagents and, consequently, a low degree of oxidation or substitution with functional groups as well as the chemical and microbiological purity of finished preparations seek to guarantee the safety of their use. The main disadvantage of using chemically modified starches in food production is the need to declare their presence on product labels by using the term “modified starch”. For consumers who do not have knowledge about these starches, the above term is difficult to understand or confusing and, as a result, contributes to the avoidance of purchasing products that contain modified starches.

## 2. Chemical Modifications of Starch for Food Purposes

Chemical modification consists of introducing new functional groups into starch molecules, which results in specific changes in the physicochemical and functional properties of starch preparations [15,16]. Such treatment of native granular starch can significantly modify its swelling capacity and solubility; the processes of gelatinization, retrogradation, and gelation; rheological properties of pastes and gels; the ability to form coatings; the resistance of pastes and gels to changing environmental conditions; and starch susceptibility to enzymatic hydrolysis [8,10,11,17,18]. Physicochemical and functional properties of modified starch preparations depend not only on the type of chemical groups introduced and the botanical origin of starch, but also on the degree of substitution and distribution of substituents in the starch molecules, i.e., the factors regulated by the modification conditions (concentration of reagents, pH, reaction time, and the presence of accompanying substances, including catalysts) [10,11,16]. The huge diversity of the functional properties of chemically modified starches as food additives means that they are used in many branches of the food industry, fulfilling various technological roles (Figure 1).

Chemically modified starches are obtained as a result of various reactions, e.g., esterification, etherification, oxidation or cationization, carried out individually or successively with mono- or bifunctional reagents [9,15,17]. The possibilities of using various modifying reagents are practically unlimited; however, due to environmental issues and health safety, only selected processes and reagents used in strictly defined amounts are permitted for modifying starches for food purposes (Table 1). Chemically modified starches used as food additives include oxidized starches, so called stabilized starches, cross-linked starches, and double chemically modified starches (Figure 2) [10,19]. Oxidized starch contains relatively bulky carboxylic groups, and the steric hindrance of these groups reduces the tendency of this starch to undergo retrogradation, thus reducing gel strength and providing viscosity stability. Oxidized starches produce aqueous dispersions of greater clarity and lower viscosity than native starch [10,15,16,17,20]. Stabilized starches result from derivatization of starch with monofunctional reagents, producing large substituent groups, primarily on the amylose chains. These groups constitute a steric hindrance for dispersed, linear starch fragments to realign and undergo retrogradation. Moreover, the inter- and intramolecular bonds within the granule are weakened, and this causes reduction in the gelatinization temperature of starch [10,15,16,17,20]. In turn, cross-linked food starch is a product of treatment of starch with bifunctional esterifying agent, introducing phosphate bonds connecting adjacent chains of starch polymers. This results in the strengthening of intramolecular starch interactions [11,21]. Consequently, the swelling of starch granules and the susceptibility of starch to enzymatic hydrolysis are reduced [10,15,16,17,20]. The chemical structures of exemplary starches modified by esterification with phosphate as a monofunctional or bifunctional reagent are shown in Figure 3. A summary of the effects of starch modification with selected chemical methods is presented in Table 2. However, it should be emphasized that in real food systems, the functional properties, including the rheological properties, of modified starch strongly depend on the presence of other components in the system and the interactions of the starch with them [7,12].

Chemically modified starches, like other food additives, must meet certain legal requirements specified for European Union countries in Commission Regulation (EU) No 231 [22]. This regulation specifies the reagents allowed for use and the permissible content of introduced functional groups and/or the residue of the reagents used (Table 1). In addition, this document specifies the description of the substance, the method of identification, and additional purity criteria, e.g., the content of sulphur dioxide or heavy metals.

## 3. Modified Starches as Food Additives—Production and Characteristics

### 3.1. Single Modified Starches

#### 3.1.1. Oxidized Starch (E 1404)

Oxidized starch is formed in the reaction of native starch with a suitable oxidizing agent. During starch oxidation, two types of reaction occur. The first reaction is conversion of hydroxyl groups of the starch monomers, primarily at C-2, C-3, and C-6 positions, into carbonyl groups followed by carboxyl groups [21], and the second reaction is starch depolymerization. Therefore, the carboxyl and carbonyl contents and the degree of depolymerization in oxidized starch are indicators of the degree of oxidation [23]. The oxidation process mainly occurs in amorphous regions of starch granules, especially on their peripheries [21]. The high content of amylose hinders oxidation of amylopectin chains in the crystalline areas since the co-crystallization of amylose with amylopectin affects the packing of double helices, which consequently limits the access of oxidizing agents to hydroxyl groups [17,20]. Moderate alkalinity of the oxidation reaction environment favours the formation of carboxyl groups. The share of carbonyl and/or carboxyl groups in the oxidized starches as well as the degree of depolymerization affects the physicochemical properties of the starch, such as swelling capacity and solubility, gelatinization time and temperature, rheological characteristics of pastes and gels, and susceptibility to retrogradation [8,10,24,25]. Oxidation causes weakening of the molecular structure of starch granules, including, destruction of amylopectin double helices, which in turn manifests in reduced temperature and reduced gelatinization enthalpy as well as in a lower swelling power of the oxidized starch [26]. Compared to native starch, the oxidized starch is lighter in colour. As a consequence of partial depolymerization, it is more soluble in water, forms pastes of lower viscosity, and produces weaker gels. Moreover, it shows a lower tendency for retrogradation, which means that oxidized starch gels are less susceptible to syneresis [10,21,27]. Limited retrogradation is due to both partial depolymerization of starch polymers and the presence of anionic groups. The presence of carboxyl groups in the oxidized starch structure makes it prone to interact with metal ions, amines, polyols, or other substances.

Although starches can be oxidized with various oxidizing agents, only sodium hypochlorite is allowed for oxidation of starch for food purposes, and the carboxyl group content in the finished preparation must not be greater than 1.1%, dwb (Table 1). Factors affecting the degree of oxidation with sodium hypochlorite include reagent concentration, pH, reaction temperature and time, the molecular structure of the starch, and its botanical origin [21,25,27]. Oxidized starches approved as food additive is used as a stabilizer, gelling agent, or thickener in confectionery products (e.g., cake fillings, powdered cake mixes), pudding desserts, or whipped cream. Moreover, this starch is used in the production of sauces, thickened soups, frozen lunch dishes, or baby food, in which it performs texturizing (binding, thickening) and stabilizing functions [8,28,29,30].

#### 3.1.2. Stabilized Starches

##### Starch Esters

Acetylated starch (E 1420)

During starch acetylation, the starch slurry is treated with acetic anhydride or vinyl acetate, which results in the substitution of hydroxyl groups of glucose monomers with hydrophobic acetyl groups [10,30]. The substituent groups are introduced in the amorphous domain of the starch granules [30]. Depending on the amount of reactant and the environmental conditions, a low, medium, or high degree of starch substitution can be obtained. Acetylated starch intended for food purposes may not contain more than 2.5% of acetyl groups, dwb (Table 1). The physical and chemical properties of acetylated starch preparations depend on the type of the modified starch, the type and amount of the reagent used, the reaction medium (aqueous or anhydrous), the pH, the reaction temperature and duration, and the presence of catalysts or other factors influencing the course of the reaction, and, consequently, the degree of substitution [8,30]. During the acetylation process, partial depolymerization of starch may also occur, which is manifested by changes in the average molecular weight of starch and a decrease in the viscosity of its pastes. Morphological changes in the structure of starch granules were also observed and included the erosion of the granule surface, the formation of pits and pores, and the tendency of granules to aggregate [10,30]. The presence of acetyl groups in the chemical structure of starch creates a steric hindrance, which loosens the starch structure and limits interactions between starch polymer chains. As a consequence, water absorption by starch granules as well as their swelling and water solubility are increased compared to native starch [9,30,31]. These changes are manifested by a lower gelatinization temperature of the modified starch [26,30]. In addition, the hindered association of starch polymer chains is manifested by restricted starch retrogradation in gels [9,10,30]. The partial leaching of amylose resulting from starch acetylation contributes to a reduction in the degree of starch crystallinity.

Acetylated starches are used in the food industry as food additives that regulate consistency and increase food stability [15,30]. The pastes formed by acetylated starch are clear and stable during heat treatment. In addition, acetylated starches have the ability to form gels, but they are not resistant to elevated temperatures, acidic environments, and mechanical forces. Therefore, acetylated starches are not dedicated to products preserved by sterilization. The main application of acetylated starches includes the production of frozen cakes, fruit confectionery fillings, fruit cakes and bakery products, protective coatings, sauces, soups, dessert concentrates, flavoured yoghurts, and thermized curds [13,28,29,30].

Monostarch phosphate (E 1410)

Monostarch phosphates are formed in the reaction of starch with phosphoric acid or salts of this acid. The chemical structure of such modified starch is shown in Figure 3a. These starches are characterized by a high degree of water binding and swelling, as well as greater solubility in cold water compared to native starch [8,32]. They form transparent pastes, which are very stable during freezing and thawing given that the modified starch practically exhibits no retrogradation tendency [32]. Monostarch phosphates do not change their structure-forming properties even after prolonged exposure to mechanical forces. However, they are sensitive to a low pH environment. To a limited extent, they have the ability to stabilize emulsion systems. Monostarch phosphates are used as food thickeners, especially in cold gelling desserts. In addition, they are used in the production of soups, vegetable sauces, ketchups, sauces for canned vegetables and meat, dressings, frozen preserves, confectionery fillings and creams, instant desserts, flavoured and thermized yoghurts [28,29].

As an alternative to the traditional method of starch phosphorylation, the use of a microwave field as a source of thermal energy was proposed [33]. It was found that the exposure to microwave radiation did not affect the type of chemical groups substituted in starch and the size distribution of molecular weights of starch esters obtained in this way. There were also no differences in the chemical and crystal structure of starch esters obtained using microwave and conventional technologies. At the same time, it was found that the modification of starch with the use of microwave radiation occurs in a much shorter time than the modification obtained with the conventional technique, which has a positive effect on the cost-effectiveness of the process [33]. In other studies, high-voltage electrical discharges have been proposed as a source of energy used during phosphorylation. It was found that such starch treatment has a positive effect on the modification efficiency, phosphorus content, and functional properties, including water absorption, solubility, and rheological characteristics of the finished preparation [34].

Octenyl succinic starches (E 1450; E 1452)

Octenyl succinic starches (OSA starches) are produced by esterification of starch in its granular form with octenyl succinic anhydride (OSA), usually in an aqueous mild alkaline medium (pH of 7.0–8.0) and at a temperature of 30 to 40 °C [35,36,37]. One of the alternative methods of producing the OSA preparations is the esterification of starch in aqueous medium catalysed by lipase at a temperature of 65 °C. The increased temperature contributes to a sufficient loosening of the starch granules and thus to a more effective penetration of the modifying agent [36]. Chemical esterification of starch with OSA can be assisted by ultrasound; microwaves; intensive stirring; or hydrothermal, mechanical, chemical, or enzymatic pre-treatment [36,37].

The introduction of hydrophobic octenyl succinate groups into the hydrophilic starch molecule gives the starch surface activity, i.e., amphiphilic properties [37], and additionally anionic properties. As a result, OSA starch becomes cold water soluble and resistant to retrogradation; it supports emulsification and perfectly stabilizes emulsions and foams [35,36,38]. Moreover, as a result of the presence of octenyl succinic groups and concomitant changes produced in the granules’ structure, OSA starches usually show a lower temperature and gelatinization enthalpy and a higher swelling power, and they form pastes of higher viscosity that are clearer compared to a native starch [31,35,36,39]. The above-mentioned properties of OSA starches imply their use as emulsion stabilizers, e.g., in milk drinks, mayonnaises, and dressings as texturizing agents (thickeners, binding agents), and in gluten-free bread as carriers and matrices for encapsulating bioactive compounds as well as raw materials for producing biodegradable films and coatings [28,36,38,40,41,42,43]. Due to instant properties, OSA starches are widely used in cold-prepared products. Starch octenyl succinates have also been shown to reduce the glycaemic response and exhibit some resistance to digestion, indicating their potential use in functional foods.

##### Starch Ether—Hydroxypropyl Starch (E 1440)

Hydroxypropyl starch (E 1440) is obtained by reacting starch with propylene oxide in the presence of a strongly alkaline catalyst, usually at 40 °C. The reaction of replacing hydroxyl groups with hydroxypropyl groups has a nucleophilic substitution mechanism [13,15]. The content of hydroxypropyl groups in hydroxypropyl starch intended for food purposes may be up to 7.0%, dwb (Table 1). The presence of hydrophilic ether groups limits the number of hydrogen bonds that join starch polymers, which weakens the internal structure of starch granules. Changes in the ordering of the structure of polymers result in their greater availability to water molecules; thus, starch becomes more hydrophilic. This results in an increase in swelling degree and viscosity of the gels formed, as well as a reduced gelatinization temperature of starch compared to native starch. In addition, it significantly reduces the ability of amylose to recrystallize, which reduces the phenomenon of retrogradation [10,13,15,16]. Hydroxypropyl starch is also stable at high temperatures and in a low pH environment. Due to the above-mentioned properties, preparations of this starch are used as thickeners in products based on water or milk that are frozen and stored in refrigerated conditions [28,29].

#### 3.1.3. Cross-Linked Starch—Distarch Phosphate (E 1412)

Starch cross-linking consists of introducing additional, stiffening cross-links to the chemical structure of starch [32]. Starch cross-linking reactions leading to distarch phosphate are carried out in an alkaline environment at a temperature of 20 to 50 °C, treating starch in suspension with phosphorus oxychloride or sodium trimetaphosphate [8,32]. The amount of phosphate in terms of phosphorus may not exceed 0.5% for potato or wheat starch and 0.4% in other cases (Table 1). In contrast to monostarch phosphate (E 1410), two hydroxyl groups of glucose units from two adjacent starch chains are esterified with one phosphate group in distarch phosphate (Figure 3b). A special feature of this type of starch is the reduced ability of starch granules to swell, therefore producing a higher gelatinization temperature, a lower tendency of starch polymers to undergo retrogradation, and a reduced susceptibility to enzymatic hydrolysis compared to the native counterpart. Depending on the degree of substitution, the viscosity of distarch phosphate pastes may be higher or lower than that of natural unmodified starch pastes [10,17,32]. Moreover, unlike monostarch phosphate, the pastes formed by distarch phosphate are not transparent. Distarch phosphates with a high degree of cross-linking form gels characterized by high rheological stability even at high temperatures, low pH, and under intense mechanical forces [16].

This modified starch is used in the production of products subjected to thermal treatment in the technological process, in particular pasteurization or sterilization. It shows a special ability to prevent thermal leakage occurring during the thermal processing of meat as well as meat and vegetable products [29]. The use of distarch phosphates in the production of meat products contributes to the improvement of the cohesiveness and consistency of products and increased yield while maintaining satisfactory sensory characteristics. Distarch phosphates also improve the stability of defrosted products, which allows maintenance of the desired consistency of ready meals. In addition, they are used as stabilizers, thickeners, binders, or carriers in products such as baby food, salad dressings, mayonnaises, soups, frozen dishes, puddings, fruit confectionery fillings, and flavoured and thermized yoghurts [28,29,32].

### 3.2. Dually Modified Starches

Dual chemical modification is a type of homogeneous dual modification, as it is a combination of two modifications of the same type. In turn, heterogeneous dual modification is a modification carried out by combining two processes of a different nature, e.g., physical and chemical processes or enzymatic and chemical processes [44]. Heterogeneous dual modification of starch is described in Section 3.3.

#### 3.2.1. Oxidized and Stabilized Starch—Acetylated Oxidized Starch (E 1451)

Acetylated oxidized starch is obtained in two stages: oxidation of starch with sodium hypochlorite at low temperature (21–38 °C) in an alkaline environment, followed by esterification with acetic anhydride under slightly alkaline conditions. Acetylation of oxidized starch improves the clarity, stability, and rheological characteristics of the gels it forms. The properties of acetylated oxidized starch are affected, apart from the botanical origin of the starch itself, by the degree of oxidation and acetylation [45]. Acetylated oxidized starch has a lower gelatinization temperature than native starch. It creates clear and highly viscous pastes stable at high temperatures and gels resistant to retrogradation. It acts as a thickening and binding agent, especially in products kept at low temperatures. In the confectionery industry, it can be used as a substitute for gelatine and acacia gum [28,45].

#### 3.2.2. Cross-Linked and Stabilized Starches

##### Phosphated Distarch Phosphate (E 1413)

Phosphated distarch phosphate is a double chemically modified starch: cross-linked and stabilized with phosphoric acid. Through cross-linking, cross-links are formed between the various chains of the starch polymers, while stabilizing phosphate functional groups are incorporated as a result of simple esterification (Figure 3c). The modified starch may contain phosphate residues (calculated as phosphorus) at levels less than 0.5%, dwb, for wheat and potato starch or less than 0.4%, dwb, for starches of other origins (Table 1) [22].

The discussed starch derivative forms transparent pastes and gels with high thickening abilities. Cross-linking reduces the sensitivity of starch to changes in temperature and pH and to mechanical factors [9], while stabilisation reduces starch retrogradation. For this reason, this modified starch is used in products that are required to be resistant to high temperatures and stable in freezing and thawing conditions. This starch is used, among others, in the production of fruit confectionery fillings, vegetable sauces, dressings, yoghurts, and desserts [4,28,29].

##### Acetylated Distarch Phosphate (E 1414)

Acetylated distarch phosphate is produced by the cross-linking of starch with phosphorus oxychloride, followed by stabilization by esterification with acetic anhydride. Phosphate content, calculated as phosphorus, must not exceed 0.14%, dwb, in potato or wheat starch derivative and 0.04%, dwb, in other starches, while the content of acetyl groups cannot be greater than 2.5%, dwb (Table 1) [22]. The E 1414 starch pastes and gels are transparent and resistant to high temperatures, acidic environments, and shear forces [46,47]. Their advantage is that they do not undergo retrogradation processes during long-term storage. Acetylated distarch phosphate is a universal binder and thickener with good texturizing and stabilizing properties [44]. Therefore, it is used in the production of mayonnaises, dressings, ketchups, vegetable and vegetable-meat sauces, and fruit concentrates. Acetylated distarch phosphate is also used in the production of fermented milk drinks, thermized curds, and low-calorie margarine [29,46,48].

##### Acetylated Distarch Adipate (E 1422)

Acetylated distarch adipate is produced by cross-linking of starch with adipic acid and stabilizing with acetyl groups derived from acetic anhydride [47]. The content of adipic groups in the obtained derivative must not exceed 0.135%, dwb, while the content of acetyl groups cannot exceed 2.5%, dwb (Table 1) [22]. Pastes and gels of the discussed starch are characterized not only by resistance to elevated temperatures and shear forces, but also resistance to weakly acidic environments [44,47]. Thus, acetylated distarch adipate is used to thicken and stabilize a wide range of food products that are subjected to thermal treatment at temperatures above 70 °C. These include, among others, ketchups, vegetable and vegetable-meat sauces, lunch concentrates, dessert powder concentrates, and baking fillings. E 1422 starch is also used in the production of mayonnaise, dressings, or fermented milk drinks [28,29,48].

##### Distarch Hydroxypropyl Phosphate (E 1442)

Stabilized cross-linked starches also include distarch hydroxypropyl phosphate, which is a starch modified by cross-linking with sodium trimetaphosphate or phosphorus oxychloride and by etherification with propylene oxide. In E 1442 starch, the content of hydroxypropyl groups must not exceed 7.0%, dwb, while the residual phosphates, calculated as phosphorus, must not exceed 0.14%, dwb, in wheat and potato starch and 0.04%, dwb, in other starches (Table 1) [22]. This starch is characterized by a reduced pasting temperature compared to the unmodified starch. It forms transparent, stable gels, resistant to mechanical processing and not subject to syneresis [47]. Distarch hydroxypropyl phosphate preparations are used in particular for the production of fruit preparations, as they give them a relatively low viscosity during thermal processing. Moreover, after cooling the finished product, they contribute to a significant increase in viscosity. In addition, they give these products a short, creamy texture that does not change during freezing and thawing. E 1442 starch is also used in the production of dairy products, including pasteurized and sterilized cream, flavoured yoghurts, and baby foods, and is also a functional additive in powdered soups or vegetable sauces [29,46,48].

### 3.3. Heterogeneously Dually Modified Starches

Due to the possibility of using very different physical processes or enzymatic reactions and many chemical reactions, the methods of producing dually modified starch preparations are practically unlimited [15]. If one of the two starch modifications used is a chemical modification and the quality criteria for chemically modified starches constituting food additives are simultaneously met (Table 1), the obtained starch preparations have the status of food additives. Examples of physical processes that can be used in combination with chemical modification include pregelatinization, high hydrostatic pressure treatment, microwave treatment, sonication, gamma radiation, pulsed electric field, extrusion and others [15,17,19]. In heterogeneous dual modifications, a very important factor influencing the functional properties of the resulting starch preparation is the sequence of applied modification processes. The first modification is aimed at increasing the effectiveness of the second modification and most often leads to the disruption of the starch granule surface and the weakening of the bonding forces of starch chains. Thus, the properties of starches modified in this way depend largely on the second modification. For example, in sonicated acetylated starch, sonication causes morphological damage to the starch granule surfaces (cracks, pores, channels), which facilitates the subsequent penetration of reactants during acetylation and more efficient substitution of starch with acetyl groups [44]. An example of a heterogeneous modification is the enzymatic hydrolysis of chemically modified starch [49,50]. The preparations produced in this manner are characterized by a reduced molecular weight of starch polymers, higher solubility, lower gelling capacity, and a reduced tendency for retrogradation compared to starch not subjected to enzymatic treatment. They form pastes of lower viscosity, but these pastes are rheologically stable [51]. The presence of stabilizing and/or cross-linking groups in the structure of the modified starch hydrolysates results in different functional properties in relation to unmodified starch hydrolysates [51,52]. For example, enzymatically hydrolysed starch octenyl succinate has the ability to reduce interfacial tension in foams and emulsion systems [32]. Far-reaching enzymatic hydrolysis of chemically modified starches leads to the production of chemically modified maltodextrins [51]. Maltodextrins obtained by enzymatic hydrolysis of chemically modified starches may be interesting stabilizers of food emulsions and modifiers of the rheological properties of the latter [52].

In the literature, the term “triple starch modification” is also used. In triple modification, the first and dual modifications prepare the starch chains, structures, and polymeric components of the starch granules for the penetration of the modifying agents during the third modification. Triple modification can be used for the synthesis of resistant starches, porous starches, and preparations with high adsorption capacities [19].

## 4. Specific Properties and Applications of Chemically Modified Starches

### 4.1. Modified Starches as Resistant Starch

Resistant starch (RS) is the part of starch and starch derivatives that resists digestion while passing through the gastrointestinal tract [53,54]. RS positively influences the functioning of the digestive tract, microbial flora, blood cholesterol levels, and the glycaemic index and assists in the control of glucose level [54,55]. Apart from the health benefits, resistant starch impacts desirable functional properties in food products, such as water-binding and swelling capacity, texturizing, enhanced viscosity, and gel-forming capacity. In addition, RS has minimal effects on the sensory characteristics of food compared to traditional sources of fibre [53,56]. There are five main types of resistant starches: RS1—starch that is physically inaccessible due to being enclosed in a plant cell wall, RS2—starch with a specific compact crystal structure that prevents enzymatic hydrolysis, RS3—retrograded starch, RS4—chemically modified starch, and RS5—amylose–lipid complexes [54,56].

RS4 includes starches substituted or cross-linked with etherifying or esterifying agents. The presence of large substituent groups and/or chemical bonds in chemically modified starch constitutes a steric hindrance for the active centre of the enzyme, which is why such starches are difficult to digest [53,54,55,56]. It has been observed that OSA starches are a source of significantly more slowly digestible and resistant starches than other modified starches, such as hydroxypropylated starch, acetylated starch, or starches cross-linked by factors used in various combinations [35]. Resistant starches of the RS4 type can also be obtained using a double modification consisting of initial starch retrogradation (by freezing) and then substitution with acetyl groups and/or cross-linking with adipic acid. Chemically modified preparations made from pre-retrograded starch are characterized by a higher degree of substitution and are more resistant to amylolysis than native starch preparations [57].

Many examples of chemically modified starches highly resistant to enzymatic digestion are described in the literature, for example starch citrate or starch malate [58,59,60]. These starches are usually obtained by modification with food-approved reagents and may be used as RS4 in the future; however, currently, due to the high level of modification, their use is not allowed in the EU. In the past, dextrins obtained from starches or acid-modified starches were referred to as food additives. Also, starch dextrinization products, resistant to enzymatic digestion, were classified by many authors as RS4.

### 4.2. Modified Starches as Fat Substitutes

Producing low-fat products that are sensory similar to full-fat products is a major challenge for the food industry. Carbohydrates, including native and modified starches, are most often used as fat substitutes [48]. Carbohydrate polysaccharides are fat mimetics, i.e., substances that organoleptically and physically imitate the properties of fat, but do not display all of its technologically important properties [61]. These polysaccharides have the ability to bind large amounts of water, which causes them to give products a texture imitating that determined by fat (e.g., softness and meltability in the mouth) [48,62]. However, they have limited technological use due to their susceptibility to high temperatures or shearing conditions. Therefore, products in which modified starches act as fat substitutes are mainly salad dressings, spreadable margarines, low-fat dairy products, low-fat mayonnaises, confectionery coatings, and meat products [28,29,61,62].

### 4.3. Modified Starches as Encapsulating Agents

Microencapsulation is the process of enclosing unstable or volatile substances, usually of a lyophilic nature, such as dyes, flavours, or other biologically active substances, in protective coatings [41]. Native starches and chemically modified starches are used in the encapsulation process [28]. Techniques that enable the encapsulation of active substances in a starch matrix include spray drying, extrusion, precipitation, and chelation. Examples of encapsulation are encapsulation of δ-limonene and betacyanins in phosphorylated starch, encapsulation of vanillin in oxidized starch, or encapsulation of volatile flavours, e.g., 1-menthol, or oils enriched with vitamins A and E in a shell made of starch octenylsuccinate (OSA) [41,62]. The latter chemically modified starch is most often used for encapsulation due to its good solubility in water, low viscosity, effectiveness in protecting active ingredients against oxygen, as well as higher production efficiency compared to other chemically modified starches [35,36]. For example, a preparation in which chlorophyll was enclosed in the OSA starch matrix was characterized by a much higher content of this active ingredient, greater antioxidant activity, and a longer shelf life than analogous preparations based on maltodextrin or gum arabic [41]. The efficiency of encapsulation of given substances by OSA starch is determined by the degree of porosity of its granules. This porosity can be increased by additional enzymatic or chemical modification of this starch [41]. Moreover, as shown by Sweedman et al. [35], OSA starch in combination with other polysaccharides, e.g., cationic chitosan, also allows very stable microcapsules to be produced. Chemically modified preparations based on waxy corn starch have very good encapsulation ability. They create stable water-oil systems; thus, they are suitable for encapsulating fat-soluble aromas, vitamins, and bioactive spice ingredients.

Porous starch preparations also have effective encapsulating properties. They are partially hydrolysed starches with a honeycomb microstructure and a well-developed specific surface, which, compared to native starch, results in a significant improvement in adsorption properties [50]. Additionally, in order to improve their properties, such as thermal stability, adsorption capacity, clarity of the produced pastes or susceptibility to enzymatic hydrolysis, they are subjected to modifications analogous to those approved for food processing purposes [50]. For example, esterification of porous starch results in a preparation with improved thermal stability, including increased resistance to freeze-thaw cycles. In turn, cross-linking of porous starches has a beneficial effect on improving the order of their structure and reduces their solubility and swelling capacity. As a result of oxidation of porous starch, the clarity and adhesiveness of the pastes it forms increases, and the gelatinization temperature and viscosity of its gels simultaneously decrease [50].

### 4.4. Modified Starches as Micronutrient Carriers

The introduction of deficient bioactive ingredients into food products is one of the trends in the production of enriched and functional food. The introduced active ingredient must be evenly distributed in the food matrix, which is a technological challenge. In the case of microelements, an effective technique is the use of a stable and effective carrier in the form of starch, especially chemically modified starch. Śmigielska et al. [63] showed that both native and modified starches, especially those with hydrophilic carboxyl groups in their structure, effectively adsorbed copper, iron, and zinc ions. In the case of oxidized starches, the efficiency of micronutrient adsorption increased with the increase in the degree of starch oxidation. In turn, starches containing more hydrophobic acetyl groups were less effective as micronutrient adsorbents. In the latter case, the cation adsorption efficiency decreased in the following order: copper > iron > zinc. The adsorption process itself did not cause significant changes in the functional properties of starch, which was proven by the example of the textural properties of traditional sweet desserts [63].

### 4.5. Modified Starches as Emulsifiers and Stabilizers

Native starches do not have emulsifying properties due to the lack of hydrophobicity and too large particle sizes, which prevent the formation of a cover around the oil globules in the emulsion. For this reason, native starches are rather used as emulsion stabilizers, and their stabilizing effect is mainly due to the ability to increase the viscosity of the continuous phase [64]. Increasing the hydrophobicity of native starch can be achieved by modifying it, e.g., by introducing hydrophobic alkenyl groups into its structure. An example of such starch, one of the most frequently used as emulsifiers, is starch octenyl succinate (OSA). The presence of hydrophobic octenyl groups in the hydrophilic starch molecule gives it surface activity, i.e., emulsifying ability, while maintaining the ability to stabilize water-oil emulsions [37]. It was found that the emulsification capacity and stability of the resulting emulsion increase as the degree of starch substitution increases [38] and that smaller granules better stabilize emulsion systems [40]. In a study by Li et al. [65], it was demonstrated that OSA-modified high amylose starch had the ability to generate optimized stabilized Pickering emulsions. The particular advantage of OSA starch is that its dispersions/pastes are almost colourless and tasteless. Moreover, unlike proteins, its emulsion-stabilizing properties are essentially independent of pH and ionic strength [35]. It has also been shown that hydrolysed OSA starch preparations, characterized by better solubility and the creation of less viscous dispersions, are even more effective emulsifiers and stabilizers [39]. In turn, nanoparticles of OSA starch preparations can be good stabilizers of Pickering-type emulsions [40,66]. The specific properties of Pickering emulsions containing OSA starches make them suitable for encapsulation of sensitive, bioactive, and/or valuable ingredients in food and pharmaceutical products [35].

Acetylated starches were also used to produce food emulsions. It was found that the stability of emulsions created using these starches increases with the degree of starch substitution, and additional cross-linking of acetylated starch contributes to the improvement of emulsion stability. The most stable model emulsions were obtained using acetylated distarch adipate. Emulsions prepared with acetylated distarch phosphate also showed similar stability [67].

### 4.6. Modified Starches in Edible Film Formulations

Starch is one of the most popular plant polysaccharides used in the production of biodegradable films [42,63,68,69]. Starch-based films are characterized by good optical, organoleptic, and barrier properties against carbon dioxide and oxygen, but less satisfactory mechanical properties [69]. Their disadvantage is high water vapor permeability, sensitivity to moisture, and consequently reduced mechanical strength [68]. Therefore, attempts have been made to overcome these limitations, e.g., by adding ingredients with hydrophobic properties (oleic acid, polyethylene glycol), co-biopolymers (e.g., high-amylose starches), or other secondary additives to the starch base [42,68,70]. Starch films with desired functional properties can also be obtained using chemically modified starches [42]. If the designed films are to have direct contact with the food product and are to be edible packaging, the modified starches used must meet the requirements for starches for food use. Oxidized starch is commonly used to prepare biodegradable films. The properties of the final product, however, depend significantly on both the botanical origin of the starch and its degree of oxidation [23]. Oxidized starches allow obtaining films with a more uniform structure, which is attributed to the partial depolymerization of starch during modification, which increases the interactions between starch and the plasticizer. Moreover, films containing oxidized starch are characterized by reduced solubility in water and water vapour permeability and are more transparent and more flexible than films made from native starch [68]. Acetylated starch has also been used to produce biodegradable films. It was shown that such films were characterized by high transparency, but the mechanical properties deteriorated. A similar effect was obtained when hydroxypropyl starch was used [68]. The mechanical strength of a polymer film can be increased by using cross-linked starches as a film-forming material. It was found that films based on such starches have much lower water content and solubility, as well as better mechanical properties than films based on native starches [42].

### 4.7. Modified Starches in Edible Coating Formulations

Edible starch coatings are increasingly used to extend shelf life and maintain quality during storage of some food products [71,72]. Starch pastes combined with an appropriate plasticizer and/or other ingredients, e.g., bioactive ones, create continuous structures suitable for direct coating of raw materials and food products [72]. Starch coatings are applied to food using various techniques: rinsing, dipping, spraying, or fluidized bed processing. These materials have many advantages, the most important of which are biodegradability, wide availability, and low price. Starch-based coatings are characterized by good rheological, optical, and sorption properties and offer appropriate gas barrier properties when used in combination with other ingredients. They are particularly useful for introducing active ingredients and additional substances into food with the possibility of their controlled release during storage, preparation, or consumption of food [71]. The limitation in using native starch as a base for coatings is the low stability of the coatings in high ambient humidity conditions. One of the solutions to eliminate this limitation is the use of chemically modified starches approved for use as food additives. Coatings made from this type of starch can be characterized by very diverse functional properties [71,72].

### 4.8. Modified Starches as Nanomaterials

From both native and chemically modified starches with the status of food additives, starch particles of nanometric sizes and with various spatial structures can be produced. Generally, starch nanostructures are defined as materials with dimensions in the range of up to 100 nm [18]. Starch nanomaterials can be classified as nanocrystals, nanofibers, nanonomicelles, nanoparticles, and nanovesicles. They can be obtained from starch, among others, using acid or enzymatic hydrolysis, anti-solvent precipitation, high-pressure homogenization, or ultrasound treatment [18,73]. The starch derivatives obtained in this way have completely different physicochemical properties than the starch raw material. The production and potential use of starch nanoparticles in food have increased recently. These products can be used to stabilize, through encapsulation, biological and synthetic compounds with antibacterial and antioxidant properties. Moreover, they can be used as stabilizers in food Pickering-type emulsion systems or as a strengthening and modifying material in food packaging [73]. It was also found that starch nanomaterials can be used as a filler to improve the thermal and mechanical resistance of food packaging. An example of nanoparticles based on chemically modified starch are OSA starch nanocrystals characterized by better emulsifying ability and the ability to stabilize capric triglycerides in water-in-oil emulsions [73].

## 5. Conclusions

Starch, as a fully renewable, biodegradable, and functional biopolymer, is widely used in various industries. Due to its multifunctionality, it is particularly appreciated in the food industry, where it is used both in its native form and after various types of modifications aimed at improving its specific functional properties. Commonly used chemically modified starches with the status of food additives must meet specific legal requirements and the highest standards in terms of their safe use. In addition to the use of chemically modified starches in food production as thickening, filling, binding, gelling, emulsifying, and stabilizing agents, the health-promoting aspect of their use is important, including resistant starch type RS4, the ability to encapsulate and act as a carrier of bioactive ingredients, and as a fat replacement. New development directions in the use of chemically modified starches include, among others, obtaining biodegradable films and edible coatings for food, as well as creating nanomaterials.

The production and use of starch and its derivatives as biodegradable products based on renewable raw materials has been increasing every year. A special industry that uses modified starches is the food industry, where, in addition to desired functional properties of these starches, appropriate safety of their use is required. The growing demand of the food industry for starch preparations with new desired functional properties forces intensification of research in this area. In addition to searching for safe reagents allowing the preparation of new chemically modified starches, an interesting direction of development is the combination of chemical modification with physical or enzymatic modification and obtaining dually or even triple modified preparations.

## Figures and Tables

**Figure 1 molecules-28-07543-f001:**
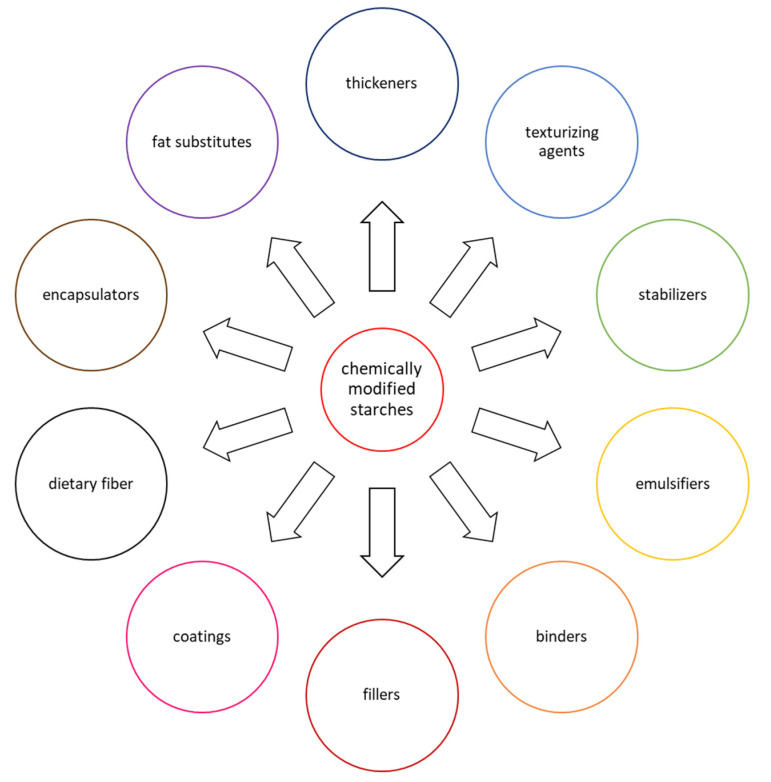
Food applications of chemically modified starches.

**Figure 2 molecules-28-07543-f002:**
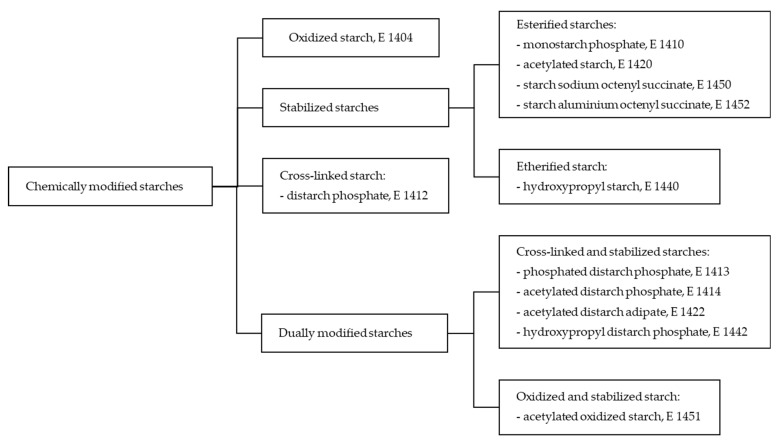
Types of chemically modified starches used as food additives.

**Figure 3 molecules-28-07543-f003:**
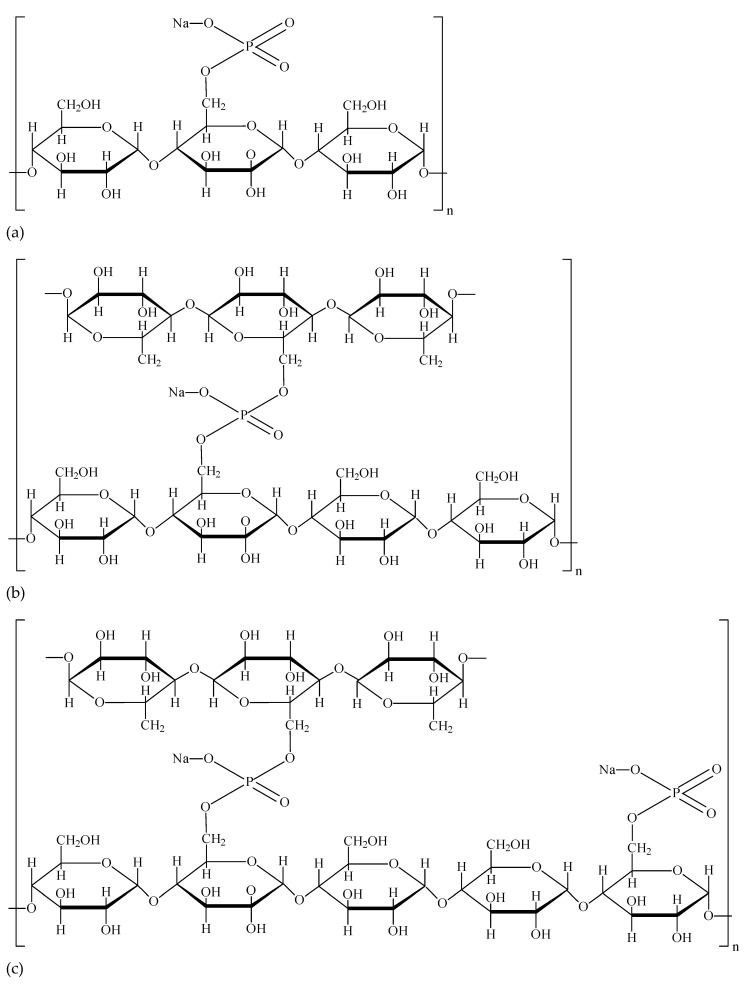
Chemical structures of starches modified by esterification: (**a**) monostarch phosphate (E 1410)—stabilized starch; (**b**) distarch phosphate (E 1412)—cross-linked starch; (**c**) phosphorylated distarch phosphate (E 1413)—cross-linked and stabilized starch.

**Table 1 molecules-28-07543-t001:** Characteristics of chemically modified starches as food additives.

Name of Modified Starch	Number	Reagent	Requirements
Oxidized starch	E 1404	Sodium hypochlorite	Carboxyl groups < 1.1% dwb
Monostarch phosphate	E 1410	Orthophosphoric acid, sodium or potassium orthophosphate, sodium tripolyphosphate	Residual phosphates (as phosphorus): <0.5% dwb for wheat or potato starches, <0.4% dwb for other starches
Distarch phosphate	E 1412	Sodium trimetaphosphate, phosphorus oxychloride	Residual phosphates (as phosphorus): <0.5% dwb for wheat or potato starches, <0.4% dwb for other starches
Phosphated distarch phosphate	E 1413	Orthophosphoric acid, sodium or potassium orthophosphate, sodium tripolyphosphate	Residual phosphates (as phosphorus): <0.5% dwb for wheat or potato starches, <0.4% dwb for other starches
Acetylated distarch phosphate	E 1414	Cross-linking agent: sodium trimetaphosphate, phosphorus oxychloride; esterifying agent: acetic anhydride, vinyl acetate	Acetyl groups < 2.5% dwb; residual phosphates (as phosphorus): <0.14% dwb for wheat or potato starches, <0.04% dwb for other starches; vinyl acetate < 0.1 mg/kg dwb
Acetylated starch	E 1420	Acetic anhydride, vinyl acetate	Acetyl groups < 2.5% dwb; vinyl acetate < 0.1 mg/kg dwb
Acetylated distarch adipate	E 1422	Cross-linking agent: adipic anhydride; esterifying agent: acetic anhydride	Acetyl groups < 2.5% dwb; Adipic groups < 0.135% dwb
Hydroxypropyl starch	E 1440	Propylene oxide	Hydroxypropyl groups < 7.0% dwb; propylene chlorohydrin < 1 mg/kg dwb
Distarch hydroxypropyl phosphate	E 1442	Cross-linking agent: sodium trimetaphosphate, phosphorus oxychloride; etherification agent: propylene oxide	hydroxypropyl groups < 7.0% dwb; Residual phosphates (as phosphorus): <0.14% dwb for wheat or potato starches, <0.04% dwb for other starches; vinyl acetate < 0.1 mg/kg dwb; propylene chlorohydrin < 1 mg/kg dwb
Starch sodium octenyl succinate	E 1450	Octenyl succinic anhydride	Octenyl succinic groups < 3% dwb; residual octenyl succinic acid < 0.3% dwb
Acetylated oxidized starch	E 1451	Oxidizing agent: sodium hypochlorite; esterifying agent: acetic anhydride	Carboxyl groups < 1.3% dwb; acetyl groups < 2.5% dwb
Starch octenyl succinate aluminium salt	E 1452	Esterifying agent: octenyl succinic anhydride; aluminium sulphate	Octenyl succinic groups < 3% dwb; residual octenyl succinic acid < 0.3% dwb

Explanation: dwb—dry weight basis. Source: [22].

**Table 2 molecules-28-07543-t002:** The effect of chemical modification of starch on selected physicochemical and functional properties.

Type of Modification	Physicochemical and Functional Properties of Modified Starch as Compared to Native Starch
Oxidation	Partial depolymerization, reduced temperatures and gelatinization enthalpy, reduced viscosity of pastes, improved thermal stability and clarity of pastes, reduced ability for gelling, reduced retrogradation, increased adhesion and ability to form films and coatings
Acetylation	Increased swelling and water absorption capacity, reduced gelatinization temperature, increased maximum viscosity, reduced retrogradation and syneresis, increased clarity of pastes, increased susceptibility to amylolytic enzymes
OSA esterification	Reduced temperature and enthalpy of gelatinization, obtaining the ability to reduce surface tension and the ability to stabilize the emulsion, obtaining the ability to encapsulate
Hydroxypropylation	Reduced gelatinization temperature, increased clarity of pastes, increased stability in freeze-thaw cycles, increased resistance to extreme environmental pH values, high temperatures and shear forces, reduced retrogradation
Cross-linking	Increased temperatures and enthalpy of gelatinization, decreased swelling capacity, increased viscosity of pastes, increased cohesiveness of pastes and gels, increased strength of gels, increased thermal stability of pastes and gels, susceptibility to amylolytic enzymes

## Data Availability

Not applicable.

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
