# Peer review of "Chemically Modified Starches as Food Additives"

_molecules, 2023, doi:10.3390/molecules28227543_

Round 1

Reviewer 1 Report

Comments and Suggestions for Authors

Starch as a renewable and multifunctional polysaccharide biopolymer and it is widely used both in the food industry and in other areas of the economy.

Chemically modified functional starch is widely used in industry. Although such starches hold the status of permitted food additives, their usage is subject to stringent regulatory oversight under relevant laws. Therefore, research and development of novel methods for these starches and associated food additives are of utmost significance.

This article provides an overview of the characteristics of chemically modified starches used as food additives, encompassing the requisites for their preparation and practical applications, carrying substantial importance.

Also, the topic of new trends in the utilization of chemically modified starches was also explored, encompassing the production of biodegradable films, edible coatings, and nanomaterials.

The conclusions consistent with the evidence and arguments presented. The text clear and easy to read, and can be recommended for publication.

Author Response

Dear Reviewer,

We are very grateful for your thorough review of our manuscript and recommendation for its publication.

With kind regards,

Lesław Juszczak

Reviewer 2 Report

Comments and Suggestions for Authors

Minor Revisions for this manuscript is suitable for publication in molecules; however there are some issues which must be clarified before publication.

Line 38: what is the relationshi between the sentence and ref N° 1!!!?

Line 59: ref N°7 seems to be the same as you current review!!!? Thus, explanation should be done?

authors should argue the need of their revue because there are several reviews dealing with this theme.

Line 153: need ref:

reaction tem perature and time, starch molecular structure, and botanical origin of the starch [****].   

Line 236: need ref.

o a more effective pen etration of the modifying agent [*****].

Line 418: need ref.

index and assists in the control of glucose level [***].

all text should be checked for references.

Author Response

Dear Reviewer,

On behalf of my co-authors, we are very grateful for discerning review of our manuscript and constructive comments. We have carefully revised the manuscript based on the reviewer’s suggestions. The changes we have made are highlighted in red font in the revised manuscript. The following part is the point-by-point responses to the reviewer’s comments.

With kind regards,

Lesław Juszczak

Minor Revisions for this manuscript is suitable for publication in Molecules; however there are some issues which must be clarified before publication.

Line 38: what is the relationship between the sentence and ref N° 1!!!?

We apologize for our mistake. The correct reference has been added.

Line 59: ref N°7 seems to be the same as your current review!!!? Thus, explanation should be done? authors should argue the need of their revue because there are several reviews dealing with this theme.

The publication mentioned by the reviewer describes chemically modified starches intended mainly for use in non-food industries, e.g. dialdehyde starch, thermoplastic starch. Moreover, this article is dated on 2011 and, thus, does not cover the latest achievements in the field of starch modifications. Moreover, it is published in Polish, therefore its availability is limited.

Line 153: need ref: reaction temperature and time, starch molecular structure, and botanical origin of the starch [****].   

The references have been added.

Line 236: need ref. o a more effective penetration of the modifying agent [*****].

The reference has been added.

Line 418: need ref. index and assists in the control of glucose level [***].

The reference has been added.

all text should be checked for references.

Whole manuscript text has been checked for references and supplemented with additional references.

Reviewer 3 Report

Comments and Suggestions for Authors

The review entitled 'Chemically modified starches as food additives' discusses on the type of chemically modified starches and their characteristics.

However, the review shows lack of significant contribution to the knowledge. It is recommended for the authors to highlight the advantages and the disadvantages of chemically modified starches as food additives and the processes involved for each modifications towards the molecular structure of the starch.

The review shows a poor organization in terms of the sub-topic arrangement and the highlights. The mechanism for each chemical modifications should be discussed by the authors. The authors are recommended to come out with more figures summarizing the concept of the review.

Comments on the Quality of English Language

The English is moderately acceptable.

Author Response

Dear Reviewer,

On behalf of my co-authors, we are very grateful for discerning review of our manuscript and constructive comments. We have carefully revised the manuscript based on the reviewer’s suggestions. The changes we have made are highlighted in red font in the revised manuscript. The following part is the point-by-point responses to the reviewer’s comments.

With kind regards,

Lesław Juszczak

General comment: The review shows lack of significant contribution to the knowledge. It is recommended for the authors to highlight the advantages and the disadvantages of chemically modified starches as food additives and the processes involved for each modification towards the molecular structure of the starch. The review shows a poor organization in terms of the sub-topic arrangement and the highlights. The mechanism for each chemical modification should be discussed by the authors.

The reviewer's suggestions have been considered. We have supplemented the text with a presentation of the advantages and disadvantages of using chemically modified starches as food additives. We have added a drawing (Figure 3) showing the chemical structures of starches modified by esterification (phosphorylation), illustrating examples of substituted starch, cross-linked starch and substituted and cross-linked starch. Moreover, additional Figure (Figure 1) representing food applications of chemically modified starches has been added. The arrangement of subchapters has been corrected and their names have been modified.

Abstract: Acceptable with slight improvement needed.

The abstract have been supplemented.

Introduction: Add-in the sources for starch Content:

Information on starch sources has been added.

  1. The resolution for Figure 1 should be enhanced (quite blurry).

The Figure 1 has been modified to improve its resolution.

  1. Suggest to add references for each type of modified starch.

The content of the paragraph has been redrafted, considering citations regarding individual starch modifications.

  1. Add references for each modified starch in Table 1.

All data presented in Table 1 are taken from the current regulation governing the specifications of chemically modified starches: Regulation (EU) No. 231/2012 of 9 march 2012 laying down specifications for food additives listed in Annexes II 652 and III to Regulation (EC) No. 1333/2008 of the European Parliament and of the Council. Off. J. Eur. Union 2012, L 83/1. This reference has been indicated in the Table 1 footer.

  1. Line 138: Missing a full-stop. Please check for all.

The text was checked and missing punctuation marks have been added.

  1. There are some imbalance paragraphs. Example: in line 147-148, the paragraph is too short. Should be combined with the following paragraph. Paragraph in line 212-215 need to be combined. Generally, a paragraph should have 4-5 sentences.

Paragraph sizes have been modified in accordance with the reviewer's recommendation.

  1. In section 3.1.1.1, add explanation on the application of oxidized starch with higher DS.

Only slightly oxidized starch (carboxyl group content < 0.1%) may be intended for food purposes, so the phrasing, suggesting that oxidized starch with a higher degree of substitution may have food applications, was incorrect. We apologize for the inaccuracy. The text has been corrected.

  1. The numberings for all sub-sections need to be checked and corrected accordingly. Example: In line 229, 3.1.1.2.2. should be corrected to 3.1.1.2.3.

Thank you for carefully checking the text. The numbering has been corrected.

  1. Every sub-topic or sub-sub-topic need to be more than 1. If not, don’t make it as a new sub-topic/sub-sub-topic. Just merge under the previous topic/sub-topic (refer to the highlighted comments in the pdf file).

The arrangement of the sub-topics and sub-sub-topics has been corrected.

  1. Line 508: The title should highlight on the “New trends of modified starch in the industries”.

The title has been modified according to the reviewer’s comment.

  1. The conclusion is poor. The conclusion should bring some insights for future studies working with the chemically modified starch/ modified starch.

The conclusions have been supplemented.

  1. Number of references must be added.

References have been added.

Round 2

Reviewer 3 Report

Comments and Suggestions for Authors

The manuscript entitled 'Chemically modified starches as food additives' has been significantly improved by the authors. The present revised form is in a good shape with provides much significant contribution towards knowledge. 

*Please delete the double word of 'cross linked' in line 20.